# Physical Engagement in Face-to-Face Interaction Is Associated with Depressive Symptoms of Interaction Partners in the Workplace

**DOI:** 10.3390/bs14111006

**Published:** 2024-10-30

**Authors:** Jong-Hyeok Lee, Nobuo Sato, Taiki Ogata, Kazuo Yano, Yoshihiro Miyake

**Affiliations:** 1Department of Computer Science, Tokyo Institute of Technology, Yokohama 226-8502, Kanagawa, Japan; ogata@c.titech.ac.jp (T.O.); kazuo.yano.bb@hitachi.com (K.Y.); miyake@c.titech.ac.jp (Y.M.); 2Happiness Planet, Ltd., Kokubunji 185-8601, Tokyo, Japan; nobuo.sato.jn@happiness-planet.org; 3Hitachi, Ltd., Kokubunji 185-8601, Tokyo, Japan

**Keywords:** face-to-face interaction, nonverbal interaction, physical engagement, depressive symptoms, organizational behavior

## Abstract

Individuals develop interdependence through interactions. The process of physical coordination during face-to-face interactions facilitates relationship formation, emotional experiences, and emotional contagion. Workplaces inherently involve organic and continuous face-to-face interactions. Examining the interpersonal association between physical movement and emotional state among employees in the workplace is crucial for understanding an organization’s emotional dynamics. This study investigated the association between patterns of physical engagement during face-to-face interactions and depressive symptoms with those of interaction partners in a real-world workplace setting. The analysis utilized face-to-face interaction data from 449 employees across ten organizations, measured using wearable devices, along with a self-reported depression scale. The results showed that employees’ average physical engagement negatively correlated with the average depressive symptoms of their interaction partners, rather than their own. The association remained robust regardless of the number of relationships or workplace type. Furthermore, the depressive symptoms of individuals who had a significant influence on interpersonal relationships within organizations negatively correlated with their and their partner’s physical engagement. Our findings have important implications for the epidemiological understanding of organizational mental health in relation to face-to-face interactions among employees.

## 1. Introduction

As mental health in the workplace has increasingly emerged as a critical issue in modern society [1,2,3], substantial progress has been made in managing and coping with stress, ranging from sleep and physical activity management to mindfulness meditation and cognitive-based therapies [4,5,6,7,8]. Meanwhile, from the perspective of supervisors and colleagues, the prevention of depression among employees in the workplace is another significant challenge. The vast majority of workplace work involves interactions among employees. Through interactions, people communicate verbal and nonverbal information, form relationships, and experience emotional contagion, such as depression [9,10,11,12]. The manner in which employees interact is closely associated not only with their productivity but also with how they encounter and manage stress [13,14,15].

The observation of interactions among multiple individuals, such as those in organizations and communities, has long focused on the interdependence that manifests in interpersonal relationships [16,17]. Interdependence through relationships affects perceptions and influences. Research indicates that individuals in direct relationships or structurally equivalent positions often share similar social perceptions [18,19], and individuals who are more central to the relationship network with better access to information and resources exhibit greater influence and performance [20,21,22]. Community-based longitudinal cohort studies have suggested that traits often perceived as individual characteristics, such as obesity, smoking habits, and depressive symptoms, can be correlated and transmitted through social networks comprising family members, neighbors, friends, and coworkers [12,23,24]. These observations are particularly significant in close and frequent relationships, rather than merely among individuals in geographical proximity. Notably, the influence was found to extend beyond immediate social ties, reaching as far as friends of friends and to friends of friends of friends in social networks.

Emotional contagion is an emotional factor that can be influenced by interactions within organizations. Emotional contagion is a complex phenomenon that occurs through automatic processes, such as unconscious mimicry and observation of others’ facial expressions, postures, behaviors, and voices, resulting in physiological feedback and mirror neuron activation [25,26]. It also involves conscious processes where individuals compare their emotions with others and adjust them according to the context [25,26]. This emotional contagion effect can have both positive and negative impacts on how groups cooperate and perform within work groups [27], leading to the formation of collective emotions [26,28]. The effects of emotional contagion can continuously influence individuals through interactions within an organization. In particular, long-lasting depressive mood states, such as depression, can have a persistent impact through interactions within the organization [29]. Observing interactions among employees can be an effective way to monitor and understand medium- to long-term mental health in the workplace.

The potential to observe interactions within organizations has expanded with the advancement of wearable sensors, enabling more quantitative and high-resolution observations and applications. Badge- or name-tag-type sensors have been developed to capture face-to-face interactions in real environments, significantly substituting for self-reported relationships [30], explaining organizational interaction patterns and efficiency [31], and even identifying isolation and low local interconnectedness related to depressive symptoms among organization members [32,33]. However, despite promising evidence from methods that estimate social relationships through the observed frequency or duration of face-to-face interactions, little is known about the dynamics occurring during these face-to-face encounters, which may contain valuable information about the qualitative aspects of the interactions, including how participants experience and influence each other throughout the encounter.

A feasible and reliable method for quantitatively characterizing face-to-face interactions is the assessment of the physical activity of participants using wearable sensors. The bidirectional association between depressive symptoms and physical activity levels is well established [34,35], and individuals with severe depressive symptoms exhibit reduced affiliative facial expressions and head movements during face-to-face interactions [36,37]. Furthermore, physical movements during face-to-face interactions are interpersonally coordinated and influence relationship formation and emotional experiences. People unconsciously mimic the behaviors, postures, and facial expressions of their interaction partners, and this mimicry is positively related to the level of smoothness of interactions and likeability assessed by the other person [38]. Movement synchrony is positively related to the rapport formed between interacting individuals [39] and affects the rapport inferred by observers of such interactions [40]. These unconscious and automatic adjustments in interpersonal interactions, such as movement, posture, and facial expressions, are facilitated by the mirror neuron system, which allows us to emotionally connect with others and interact smoothly. This has been suggested as a potential mechanism for the contagion of emotional states such as depression [11]. Additionally, the direction of emotional contagion and the degree of tendencies of movement synchrony are influenced by personal traits such as physiology, personality, and prosocial orientation [11,41]. Interpersonal physical and emotional interactions have also been directly associated with depression. The dysfunction of the mirror neuron system observed in patients with major depressive disorder is associated with difficulties in emotional processing [11]. Incongruent levels of nonverbal involvement behaviors such as gestures and head movements during face-to-face interactions may induce the recurrence of depression, which is associated with an increased occurrence of stressful events [42]. In summary, physical movements during interactions are closely related to the emotional states of individuals and their interaction partners, facilitating emotional contagion and induction between individuals. Observing physical movements during face-to-face interactions can provide valuable information for assessing the emotional states of individuals and those around them.

In this study, we investigated the association between employees’ physical movements during face-to-face interactions and their emotional states with those of their interaction partners in the workplace. We observed face-to-face interactions among 449 employees from 10 organizations in various occupations in real-world workplace settings. For observation, a name-tag-shaped wearable device [43] designed to be comfortable in workplace environments was used. The wearable device detected face-to-face interactions with other employees wearing it and measured physical movements by tracking the frequency of upper-body movements, referred to as body rhythm. Specifically, we focused on the increased body rhythm during interactions, considering all possible physical movements such as head nodding, gestures, and synchronization that occurred during interactions. Increased body rhythm during face-to-face interactions, referred to as physical engagement, was quantified by subtracting the average body rhythm before the interaction from the average body rhythm during the interaction. To assess employees’ emotional states, we used a self-reported depression scale (Center for Epidemiologic Studies Depression Scale; CES-D) [44], which is widely used in epidemiological studies of the general population [12,32,33] and in screening depression. For detailed descriptions of the data, please refer to Section 2.

Using the data, we comprehensively investigate the correlations between physical engagement and depression scores associated with employees and their partners during face-to-face interactions. Additionally, we examine the robustness of the results to determine whether they vary according to the number of relationships or workplace settings. Furthermore, we investigate the characteristics exhibited by individuals who have a significant influence on observed face-to-face interactions within the workplace and discuss how these observed characteristics can be understood in conjunction with the findings on employees. Through these investigations, we present evidence for interpersonal associations between physical engagement and mental health in the workplace and discuss how these findings may relate to and impact organizational mental health.

## 2. Materials and Methods

### 2.1. Description of Data

The dataset was collected by the Hitachi World Signal Center in 2009 and comprises data from 470 employees across ten organizations in Japan. The target organizations spanned a variety of occupations, including financial systems, public systems, call centers, IT services, personnel applications, industrial equipment, medical equipment, and insurance. Interactions among employees were observed for more than five days in all organizations, with a mean observation period of 11.5 days (*SD* = 7.2).

All employees in the target organizations were informed about the purpose of the measurement and the use of the data. The measurement was conducted only with participants who provided informed consent. All workplaces and employees in the dataset were fully anonymized to prevent the identification of specific individuals or organizations. This study utilized the same dataset as Lee et al. [33].

### 2.2. Wearable Device

The wearable device [43] was designed to be compact (73 × 98 × 9 mm, 62 g) to ensure that employees could use it comfortably in a real-world workplace setting. It was also designed as a name-tag type, which is a familiar format for companies. The name-tag type device is advantageous for measuring face-to-face interactions and for measuring the movement of the upper body, which is crucial for interactions. Furthermore, it was engineered to reduce power consumption and memory usage, allowing it to be used throughout the workday without charging (approximately 24 h). This comfortable design with low power consumption is advantageous for measuring behavioral trends observed over relatively long periods in large groups apart from accurately capturing individual movements.

### 2.3. Face-to-Face Interaction Data

Face-to-face interactions among employees were observed using a name-tag-type wearable device, which was designed to be comfortably worn in workplace environments. During the measurement period, participants wore their wearable devices upon arriving at the workplace and proceeded with their usual work activities. The device uses infrared modules to detect other devices within a 2 m range and a 120-degree circular sector in front of the device and communicates the device owner’s ID. Face-to-face interaction occurrences were determined per minute based on these records.

#### 2.3.1. Body Rhythm

The device uses an accelerometer to measure physical movement. The 3-axis accelerations near the upper body, measured at a resolution of 50 Hz, were processed using the Euclidean norm. The number of points at which this norm acceleration value crossed its mean was counted every 10 s, and the average of six measurements (1 min) was calculated. In this study, we refer to this frequency as the body rhythm.

Body rhythm is useful for recording periodic and patterned physical activities, along with measuring activities with strong movements in one direction. While body rhythm alone cannot predict exact movements accurately, the following are typically observed: running (over 4 Hz), excited discussion or rushed walking (3–4 Hz), walking, talking with dynamic gestures (2–3 Hz), talking or typing (1–2 Hz), web browsing or listening (0–1 Hz), sleeping, or thinking without movement (0 Hz) [45]. Although the physical engagement value in the results uses the same unit of Hz, it is difficult to interpret it directly as Hz because it deals with the average body rhythm value during face-to-face interaction time and is the value obtained by subtracting the average body rhythm value that contains contextual information immediately before the interaction.

#### 2.3.2. Data Preprocessing

In the detection of face-to-face interactions using name-tag-type wearable devices, misalignments can occur due to body orientation, posture, or movement, potentially causing missed detections. Consequently, short intervals between recorded face-to-face interaction events may represent misalignment errors. To compensate for missing data due to this misalignment, we considered face-to-face interactions that occurred within intervals of less than five minutes as continuous interactions.

Interactions observed with intervals of less than five minutes between two individuals accounted for 38.8% of the total recorded interactions. We determined that, given that our observations spanned multiple days (five or more), treating these intervals as continuous interactions, rather than new ones, was more appropriate. Additionally, our previous study [33] confirmed that interval adjustments between one and ten minutes had no significant effect on the structural centrality characteristics of the network, indicating that this correction method operates robustly without disproportionately emphasizing interactions between specific individuals.

To eliminate unreliable data owing to insufficient records, we excluded data from participants with less than five hours of device usage or less than one hour of total face-to-face interaction time, resulting in a final sample of 449 employees. For detailed Appendix A on these criteria, please refer to our previous work [33].

#### 2.3.3. Physical Engagement

Physical engagement is defined as an increase in body rhythm due to face-to-face interactions as follows:(1)$${PE}_{i}(k)={\bar{{BR}_{i}(t)}}_{t\in T_{k}}-{\bar{{BR}_{i}\left(t\right)}}_{t\in \hat{T_{k}}}$$
where ${PE}_{i}(k)$ represents the physical engagement of individual i in face-to-face interaction k, ${BR}_{i}(t)$ denotes the body rhythm of i at time t, $T_{k}$ is the set of interaction k times, and $\hat{T_{k}}$ is the set of interactions 30 min preceding interaction k. The overbar denotes the mean of the respective values.

The subtraction of ${\bar{{BR}_{i}\left(t\right)}}_{t\in \hat{T_{k}}}$ aims to establish a baseline for body rhythm preceding the interaction k. However, if there is an interaction with the partner of interaction k within the baseline window, this definition of the baseline is compromised. To address this issue, we integrated such instances into a single interaction, eliminating occurrences of the partner during the baseline. Nevertheless, this method could not fully exclude the influence of partners other than the partner of interaction k. We note that the 30 min baseline window preceding each interaction contained an average of 4.13 (median of 1) records of interactions with other individuals. Considering that the choice of baseline window span could affect the results, we demonstrated the robustness of our findings by confirming that the 30 min setting did not significantly impact the main findings of this study within spans ranging from 10 to 60 min (see Appendix A).

Furthermore, there is an increasing trend in body rhythm around 10 min before and after interactions (see Appendix A). This could be attributed to movement or preparation before the interactions. However, this could also be attributed to greetings, conversations, or physical activities influenced by the mutual awareness that occurs when the sensors are not properly aligned to detect each other. As the possibility of intermittent sensor misalignment between interaction logs was accounted for in the data preprocessing, we opted to incorporate the 10 min window before and after the interaction into $T_{k}$ rather than excluding it, while simultaneously shifting the baseline $\hat{T_{k}}$ to 10 min prior to its original time point.

### 2.4. Depression Scores

Depressive symptoms were evaluated using the CES-D scale [44]. The CES–D is a self-report depression scale developed for screening depression in the general population. The scale consists of 20 items phrased in simple and easily comprehensible language, making it accessible for the general public to respond. Of these 20 items, 16 address negative experiences (e.g., “I felt depressed”, “People were unfriendly”) and 4 focus on positive experiences (e.g., “I was happy”, “I felt I was just as good as other people”). Participants rate how often they experience each symptom over the past week on a scale of 0 (rarely or none of the time) to 3 (most or all of the time). The total scores range from 0 to 60, with higher scores indicating more severe depressive symptoms. The reliability and validity of the CES-D are well established [44,46,47,48,49], and it has been utilized to investigate the epidemiology of depressive symptoms across diverse populations. It has been utilized in research on the contagion of depressive symptoms through social networks among community populations over 20 years [12], as well as in studies related to loneliness, where it served as a representative measure for cross-sectional assessments [32]. In this study, the internal consistency of the CES-D scale was assessed using Cronbach’s alpha. The analysis yielded a value of 0.87, indicating good reliability among the items. We note that we do not have CES-D details of all participants, and 371 subjects of 449 were used to calculate this Cronbach’s alpha.

In our study, the CES-D was measured once for each employee and used as a score to represent the severity of the employees’ depressive symptoms. Requests to respond to the CES-D assessment were communicated to all participants through organizational secretariats. Participants provided responses on paper or Excel files, including their names and wearable device IDs. Completed assessments were sent directly to the principal investigators, bypassing the secretariats to prevent potential response bias that could arise if participants believed their superiors or secretariat staff might see their answers. The investigators removed names from the responses, retaining only device IDs, and securely disposed of the original documents containing names as confidential information.

### 2.5. Characteristics of Face-to-Face Interactions of an Employee

The features of each observed face-to-face interaction are as follows:(2)$$F_{i,j}(k)=\left[\begin{array}{cccc} {PE}_{i}(k) & {PE}_{j}(k) & {DS}_{i} & {DS}_{j} \end{array}\right]$$
where $F_{i,j}(k)$ represents the features observed in interaction k between employees i and j, ${PE}_{i}(k)$ represents the physical engagement of individual i in face-to-face interaction k, ${DS}_{i}$ represents the depression score of individual i, and ${PE}_{j}(k)$ and ${DS}_{j}$ represent *j*’s features in the same manner.

We define the characteristics of face-to-face interactions of employee i by averaging all observed $F_{i,j}(k)$ values observed from employee i, as follows:(3)$${\bar{F_{i,p_{i}(k)}(k)}}_{k\in I_{i}}=\left[\begin{array}{cccc} {\bar{{PE}_{i}(k)}}_{k\in I_{i}} & {\bar{{PE}_{p_{i}(k)}(k)}}_{k\in I_{i}} & {DS}_{i} & {\bar{{DS}_{p_{i}(k)}}}_{k\in I_{i}} \end{array}\right]$$
where $p_{i}(k)$ represents the partner of employee i in interaction k, $I_{i}$ represents the group of interactions of employee i, ${\bar{F_{i,p_{i}(k)}(k)}}_{k\in I_{i}}$ represents the characteristics of face-to-face interactions of employee i. The overbar notation denotes the mean of the respective values. Note that for ${DS}_{i}$, because it was measured once, the average was the same as that of a single measurement.

### 2.6. Statistical Analysis

A one-tailed paired *t*-test was conducted to examine the increase in body rhythm during face-to-face interactions compared with the baseline in Section 3.1 (Figure 1b). The distribution had a mean of 0.26 and a median of 0.24. Its skewness (0.0090) was close to zero, indicating near-symmetry, while its kurtosis (0.97) was lower than that of a normal distribution (3), suggesting a slightly flatter shape. Q-Q plot analysis (see Appendix A) showed data points closely aligned with the ideal normal distribution line in the central region, with minor deviations only at the extremes. This indicated a distribution very close to normal, but with slightly thicker tails. Overall, the variable’s distribution sufficiently approximated normality, validating the use of parametric statistical methods like the *t*-test.

Pearson’s correlation analysis was conducted to examine the association between physical engagement and depression scores among employees and their face-to-face interaction partners. Pearson’s correlation coefficient was used to numerically interpret the strength, direction, and significance of the relationship between the two continuous variables. We observed that the data points in the scatter plot of the main results (Figure 2b,d) exhibited a linear pattern, and we confirmed that the results were robust in response to outliers or nonlinearity by using Spearman’s correlation coefficient (see Appendix A). Two-tailed tests were performed for correlation analysis.

Statistical significance was set at *p* < 0.05 for all analyses. Statistical analyses were performed using Python (version 3.11, available at http://www.python.org) and SciPy (version 1.10.1) [50]. We used Seaborn (version 0.12.2) [51] for data visualization and illustrated the confidence intervals through bootstrap resampling. Bootstrap resampling was used to depict the 95% confidence intervals for the linear regressions in Figure 2a–d and Figure 3. In the bootstrap resampling process for linear regressions, data were randomly resampled with replacements of the same size as the data size. In this study, we set the number of resamples to 1000. For each resampled dataset, regression was performed using the least squares method. The confidence interval calculated using the predicted Y values from each resampling is shown as the shaded area. Wider confidence intervals indicated sparse data in that region and were included to aid in visual assessment along with the regression line.

## 3. Results

### 3.1. Physical Engagement During Face-to-Face Interactions

In face-to-face interactions, the body rhythm tends to increase. Figure 1a shows an example of an employee’s body rhythm observed during a face-to-face interaction (Figure 1a; interaction) and in the preceding 30 min (Figure 1a; baseline). The baseline window contains the employee’s base body rhythm pattern before the interaction. This includes personal, emotional, and work-related contexts that are less relevant to the interaction. In the interaction window, body rhythm patterns induced by interactions such as nodding, gesturing, synchronization, or arousal level are added. In other words, the increased body rhythm in the interaction window includes information on interaction-induced movements. Figure 1b shows the distributions of mean body rhythms at baseline and the interaction window for all observed face-to-face interactions. To analyze the changes in body rhythms between the baseline and interaction windows, we conducted a one-tailed paired t-test. The results reveal a significant increase (*t* (53,761) = 99.60, *p* < 0.001) in body rhythms during the interaction window (*M* = 1.36, *SD* = 0.59) compared with the baseline window (*M* = 1.10, *SD* = 0.60). This suggests that face-to-face interactions significantly increase participants’ body rhythms. The increased body rhythms suggest that physical engagement, quantified as the difference in mean body rhythm between the interaction and baseline windows (see Section 2 for details), captures the interaction-induced body movements during face-to-face interactions.

**Figure 1 behavsci-14-01006-f001:**
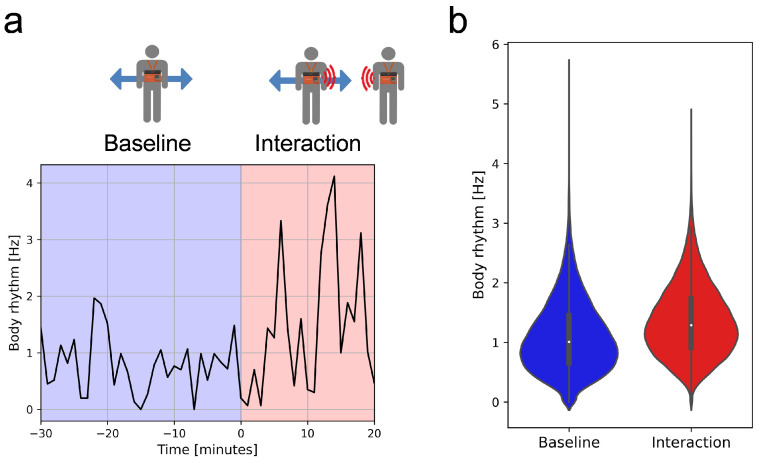
Body rhythms during face-to-face interaction. (**a**) Example of body rhythm of an employee during face-to-face interaction and the preceding 30 min. The blue area represents the body rhythm during the 30 min preceding the face-to-face interaction (baseline), whereas the red area depicts the body rhythm during the interaction. (**b**) Distribution of mean body rhythms during baseline and interaction windows for each observed face-to-face interaction. The shape of the distribution represents the kernel density estimation of the underlying distribution, which is proportional to the number of data points for each body rhythm. The white dot indicates the median, and the thick black bar represents the interquartile range.

**Figure 2 behavsci-14-01006-f002:**
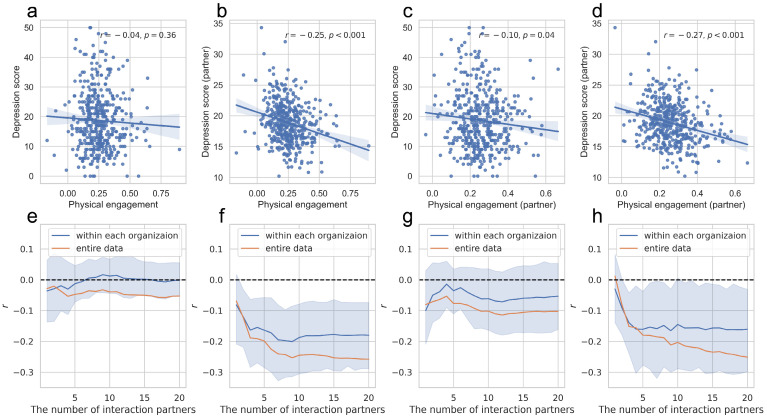
Correlation between physical engagement and depression scores of employees and their partners. (**a**–**d**) represent the correlations between (**a**) employees’ physical engagement and their depression scores in face-to-face interactions, (**b**) employees’ physical engagement and their interaction partners’ depression scores, (**c**) interaction partners’ physical engagement and employees’ depression scores, (**d**) interaction partners’ physical engagement and their own depression scores. The line represents the linear regression line, and the shaded area indicates the 95% confidence interval of 1000 bootstrap resamples. (**e**–**h**) examine the correlations in (**a**–**d**) after controlling for the number of primary interaction partners, in the entire data (orange line) and within each organization’s data (blue line). The blue-shaded area indicates the 95% confidence interval of 1000 bootstrap resamples.

### 3.2. Association of Physical Engagement with Depressive Symptoms of Interaction Partners

We investigated the characteristics of the interactions observed among the employees. For each employee, we calculated two average values of physical engagement, one for the employee and one for their interaction partners, averaging across all recorded interactions. Depression scores were calculated using the same method, noting that an individual’s depression score remained constant across interactions as it was measured once per person. For a detailed description of these calculations, please refer to the Methods section. Subsequently, we examined the correlations between physical engagement and depression scores of employees and their partners.

The correlation analysis shows no correlation between employees’ average physical engagement and their depression scores (*r* = −0.24, *p* = 0.36; Figure 2a). Meanwhile, an employee’s average physical engagement demonstrates a significant negative correlation with the average depression scores of the partners in their face-to-face interactions (*r* = −0.25, *p* < 0.001; Figure 2b). Similarly, the average physical engagement of partners in face-to-face interactions negatively correlated with the depression score of an employee (*r* = −0.096, *p* = 0.041; Figure 2c) and with the depression score of the partners in an employee’s face-to-face interactions (*r* = −0.27, *p* < 0.001; Figure 2d). These results suggest a negative interpersonal and collective association between physical engagement and depressive symptoms among employees. The interpretation of each correlation is discussed in the Discussion section.

Additionally, we investigated the robustness of those associations. First, we controlled for potential effects due to the variance in the number of interaction partners each employee had. We controlled for the number of primary interaction partners of each employee, ranked them by interaction frequency, and examined the resulting correlations. Only original partners were included when the adjusted number of partners exceeded the original number. The numbers of partners observed among employees show the following statistics: *M* = 24.06, *SD* = 10.11, *min* = 4, *max* = 60. Each correlation coefficient remained consistently stable for approximately 10 partners (Figure 2e–h). A significant negative correlation exists between the two partners for the correlation between physical engagement and depression scores (partner) (*r* = −0.12, *p* = 0.008; Figure 2f, orange line) and between three partners (*r* = −0.15, *p* = 0.001; Figure 2h, orange line) for the correlation between physical engagement (partner) and depression scores (partner). Meanwhile, there is a tendency for the correlation between physical engagement (partner) and depression scores (partner) to continuously increase as the number of included partners increases, following an initial steep rise (Figure 2h orange line). In Section 4, we discuss how this increasing pattern may reflect the characteristics of key partners and interaction groups. Second, we verified whether these correlations were observed at each workplace. The blue lines in Figure 2e–h show the mean of the observed correlation coefficients while controlling for the number of partners. The blue-shaded area represents the 95% confidence interval of the correlation coefficients observed in each organization. To address the small sample size, we computed confidence intervals for the correlation coefficients using bootstrap resampling with 1000 iterations, providing a more robust estimate of the true correlation coefficient while mitigating limitations associated with the limited sample size. The resulting intervals indicate the range within which we can be 95% confident that the true correlation coefficient lies.

The results indicate that in each organization, the correlations between physical engagement and depression scores (partner) demonstrate a significant negative association (Figure 2f, blue line), while the correlations between physical engagement (partner) and depression scores (partner) also show a significant negative association (Figure 2h, blue line). These results demonstrate the robustness of the negative correlation between physical engagement and depression scores (partner) and the negative correlation between physical engagement (partner) and depression scores (partner).

### 3.3. Key Interaction Partners and Their Characteristics

The observed negative interpersonal associations were examined in real-world workplace settings where the proportion of partners in an employee’s face-to-face interactions was not controlled. Therefore, the characteristics of employees who are frequently identified as primary interaction partners by colleagues become more significant for these associations. In this study, we refer to these individuals as key interaction partners.

To identify the key interaction partners, we selected the top three interaction partners for each employee based on frequency. “Three” is the number at which our key findings (Figure 2f) became consistently observable when we controlled for the number of primary partners, and we used this as the criterion for the number of key individuals for each employee. We aggregated the number of times each employee was selected as a key interaction partner within their workplace. And we investigated the characteristics of three groups—high, middle, and low—based on the frequency of their appearance as key interaction partners, reflecting the importance of their interactions within the organization. The tertile method was used for this grouping. While this method does not provide clear boundaries that distinctly separate the groups, it is useful for comparing overall differences in importance by dividing the population into three groups of similar size. The upper group included those with values above the tertile points: high (*n* = 115), middle (*n* = 130), and low groups (*n* = 204).

Figure 3 shows the results of the correlation analysis for the high, middle, and low groups. As a result, in the high group, referred to as key interaction partners, all correlations between physical engagement and depression scores are significantly negative, within individuals and between partners: correlation between physical engagement and depression score (*r* = −0.20, *p* = 0.03; Figure 3a), physical engagement and depression score (partner) (*r* = −0.23, *p* = 0.01; Figure 3b), physical engagement (partner) and depression score (*r* = −0.22, *p* = 0.02; Figure 3c), physical engagement (partner) and depression score (partner) (*r* = −0.24, *p* = 0.009; Figure 3d). This suggests that a prominent entanglement of physical engagement and depression scores was observed in intrapersonal and interpersonal relationships among key interaction partners.

**Figure 3 behavsci-14-01006-f003:**
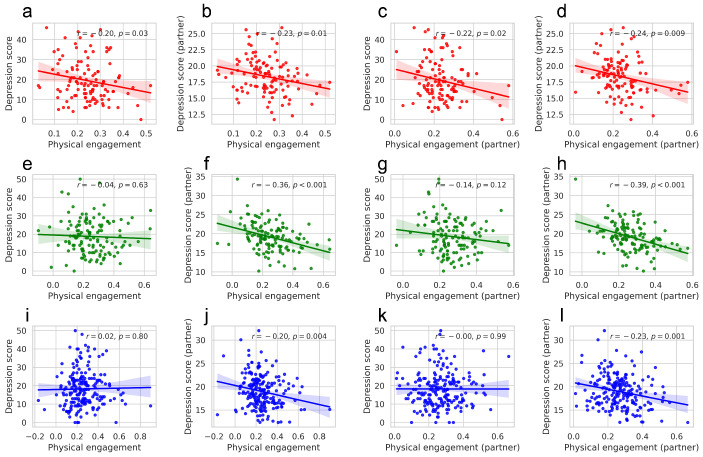
Correlation analysis in high (**a**–**d**; red), middle (**e**–**h**; green), and low groups (**i**–**l**; blue), categorized based on the number of times an employee is identified as a primary interaction partner by other employees in the workplace. In each group, the correlations are shown from left to right in the following order: physical engagement and depression, physical engagement and depression (partner), physical engagement (partner) and depression, physical engagement (partner) and depression (partner). The line represents the linear regression line, and the shaded area indicates the 95% confidence interval of 1000 bootstrap resamples.

Meanwhile, significant negative correlations were observed between physical engagement and depression scores (partner) in the high (*r* = −0.23, *p* = 0.01; Figure 3b), middle (*r* = −0.36, *p* < 0.001; Figure 3f), and low groups (*r* = −0.20, *p* = 0.004; Figure 3j). Moreover, significant negative correlations were observed between physical engagement (partner) and depression score (partner) in high (*r* = −0.24, *p* = 0.009; Figure 3d), middle (*r* = −0.39, *p* < 0.001; Figure 3h), and low groups (*r* = −0.23, *p* = 0.001; Figure 3l). This suggests that these associations are consistent characteristics of the interactions observed in organizations. The interpretation of these findings is presented in Section 4.

## 4. Discussion

While physical coordination during face-to-face interactions has received significant attention in relation to relationship formation and the sharing, feeling, and transmission of emotional states, little is known about how such interpersonal physical and emotional associations manifest in organizations that inherently involve interactions among members. We found that employees’ average physical engagement had a negative association with their interaction partners’ average depression scores rather than their own depression scores. This interpersonal correlation demonstrated robustness when controlling for the number of primary interaction partners and exhibited consistency across various workplace settings. Furthermore, for individuals who were frequently identified as primary interaction partners, we found a negative correlation between physical engagement and depression scores, which was evident in the interpersonal and intrapersonal contexts.

Several points warrant discussion regarding the interpersonal relationships between physical engagement and depression scores. First, the negative correlation observed between employees’ average physical engagement and the average depression scores of their interaction partners (Figure 2b) is consistently confirmed under various conditions: when controlling for a few key partners (Figure 2f, orange line), across different organizations (Figure 2f, blue line), and among the high, middle, and low groups defined by the importance of key partners (Figure 3b,f,j). However, a negative correlation between physical engagement and depressive symptoms was not consistently observed (Figure 2a). The significant negative correlation between an individual’s physical engagement and depressive symptoms was confirmed only in high groups defined by the importance of key partners (Figure 3a). This suggests that for individuals in significant organizational positions, such as those in superior positions, there could be a bidirectional link between their own physical engagement and depressive symptoms. Conversely, individuals with less significance in organizational interactions would not experience substantial impacts on their own depressive symptoms related to interactions. Nonetheless, the consistently observed negative correlation between employees’ average physical engagement and the average depression scores of their interaction partners suggests that the manner in which employees physically engage in face-to-face interactions would be more closely related to the emotional states of those they interact with. While causal interpretations are limited, we can consider several possibilities: frequent interactions with depressed employees may lead to decreased physical engagement, deficient physical engagement might negatively impact partners’ experiences, or structural and contextual factors might be involved. We further discuss potential explanations in the latter part of the discussion section.

Second, the association between an employee’s depression scores and their partners’ physical engagement (Figure 2c) was weaker than that between employees’ physical engagement and their partners’ depression scores (Figure 2b). This difference may be attributable to the issue of resolution, as depression scores were not measured for each individual interaction. For low-resolution observations, the averaged values may be more advantageous for explaining the trends. The introduction of quantifiable emotional state indicators for each interaction is expected to yield a more precise resolution. Another possible explanation is individual differences. For instance, key interaction partners in the workplace exhibited a stronger correlation between depression scores and the partners’ physical engagement (Figure 3c). Research that considers personal traits can provide a deeper understanding of these associations.

Third, the significant negative correlation between average physical engagement and partners’ depression scores in face-to-face interactions (Figure 2d) presents an intriguing paradox. The correlation between physical engagement and depression scores, absent at the individual level, emerges when individuals are included as interaction partners for others. One possible explanation is that the correlation may be the result of overrepresentation of the characteristics of key interaction partners. Indeed, we found a significant interpersonal correlation between physical engagement and depression scores among key interaction partner groups (Figure 3a). The intrapersonal correlations of key interaction partners, frequently included in other employees’ interaction partners, might have contributed to the observed negative correlation (Figure 2d) and the initial steep rise in the correlation coefficient when adjusting the number of primary partners (Figure 2h). Another possible explanation is that the results may include the characteristics of the group in which the employees frequently interacted. The interpersonal correlation between physical engagement and depression scores could manifest as a group correlation among those who frequently interacted with each other. The gradually increasing correlation coefficient observed after adjusting for the number of primary partners might capture this collective structure (Figure 2h). Future studies that consider the structure of these interactions could deepen the understanding of this potential explanation.

Our findings can be interpreted in relation to previous studies that have highlighted the importance of physical engagement in face-to-face interactions. Individuals with depression tend to show a decreased frequency of nonverbal behaviors and experience difficulties in processing nonverbal communication signals [11,36]. Deficits in nonverbal interactions are associated with the frequency of unsatisfactory and stressful interpersonal events and are potentially related to depression-inducing mechanisms [42]. Indeed, we confirm that among employees with a high proportion of interaction with others, their depression scores negatively correlated with the physical engagement of their interaction partners, and even with their own physical engagement (Figure 3a,c), which is not observed in employees with medium or low interaction frequencies. People tend to synchronize and coordinate their behavior with others, such as by mirroring postures or using gestures. This facilitates smoother, more effective communication and strengthens emotional connections [38,39]. Low physical engagement in interactions suggests that this behavioral coordination would not be activated, potentially leading to more easily triggered negative emotions in interpersonal relationships. This suggests a potential mechanism where cumulative deficiencies in nonverbal communication during interactions, both from oneself and others, could contribute to depressive symptoms, while conversely, active engagement in nonverbal communication could serve as a protective factor against depression.

Face-to-face interactions could also reflect the dynamic emotional interplay between participants. Automatic mimicry in facial and bodily expressions during interactions has been suggested as a channel for emotional transmission and reception [11]. Our findings indicate that employees who frequently interact with individuals showing high depressive symptoms exhibit low physical engagement (Figure 2b) and experience low physical engagement from their partners (Figure 2d). A plausible explanation for these findings is that physical engagement could be adjusted in response to the depressive symptoms of interaction partners, and also shared among them. Meanwhile, for key interaction partners who frequently engage in interactions within the organization, their physical engagement (Figure 3a) and that of their interaction partners (Figure 3c) show a negative correlation with the key interaction partner’s depressive symptoms. This correlation could result from key interaction partners being influenced by their interaction partners, but conversely, it could also be a result of behavioral (Figure 3c) and also mental (Figure 3b) adjustment occurring in their interaction partners in response to the key interaction partners’ depressive symptoms and physical engagement levels. This implies that behavioral mimicry from highly stressed, frequently interacting individuals can have a substantial impact on the physical engagement among employees, potentially influencing the replication of depressive emotions.

Our findings have significant implications for workplace applications. Unlike conventional individual-focused approaches to workplace mental health, our study highlights the potential for observing and intervening in workplace mental health through face-to-face interactions. The importance of high-quality interactions among employees and the creation of environments conducive to their emergence have received significant attention in organizational management, particularly from the perspective of driving task performance and innovation [52,53,54]. The positive impact experienced through interpersonal interactions has primarily been assessed through the concept of “energy” [52,54]. This energy is linked to positive psychological resources and the ability to cope with stress [55]. Individuals who struggle to manage stress may potentially lower the energy levels of those they interact with. Similarly, our study confirmed a tendency for decreased physical engagement in people who interact with individuals exhibiting high levels of depressive symptoms. While there is a distinction between cognitive and physical energy, future research could explore the relationship between these two types of energy.

Finally, while social contagion theory has treated depressive symptoms as transmissible through interactions and social networks [11,12], the mechanisms and observability of depression contagion in real-world settings have remained elusive. This study demonstrates that within organizations, as interactive groups, employees’ depressive symptoms can be observed as reduced physical engagement during interactions, which could also correlate with their own depressive symptoms. This observed correlation with surrounding individuals through interactions provides evidence supporting the social contagion theory of depressive symptoms, also suggesting a potential mechanism of physical and mental interplay in how depressive symptoms spread within organizations.

Despite these possible interpretations and their applications, definitive causal interpretations are limited. Physical engagement during interactions may have influenced employees’ depressive symptoms or vice versa, or they may have occurred simultaneously, or they could have been the result of other factors reflected in each characteristic. Future studies should investigate these mechanisms by using interventions or longitudinal designs. Second, depressive symptoms are influenced by various factors and not just workplace interactions. Future studies could increase the resolution of observations by narrowing the scope of symptoms to the workplace or by assessing items that may be related to physical engagement during interactions (e.g., stressful interpersonal events). Third, the CES-D scale was used once and used as a representative value of an employee’s emotional state. As discussed earlier in the Discussion section, this could be a cause of the low resolution in estimating emotional states. Nevertheless, the CES-D scale has been confirmed to have adequate test–retest reliability in several studies when the same test is repeated on the same participants after two weeks or longer periods [44,46,47,48,49]. This suggests that our observations can address a person’s depression-related emotional state, which tends to be significantly stable over a long period. Fourth, there are limitations in using physical engagement as an indicator of nonverbal communication. Previous studies of nonverbal interactions have predominantly emphasized physical synchrony among participants [11,38,39,40,41]. Physical engagement, introduced to assess the level of bodily involvement during interactions, may serve as a pragmatic measure of nonverbal communication activities in uncontrolled real-world settings. However, the omission of contextual information such as synchrony may limit the potential for more detailed observations and interpretations. Fifth, although this study encompasses a large number of employees across various organizations, the data do not contain personal demographic information (e.g., age and sex), individual differences such as specific personality traits (e.g., extraversion, neuroticism) and physiological factors (e.g., cortisol levels), and contextual workplace information such as job positions and affiliations. For example, extraverts are more expressive and responsive to positive emotions, while those high in neuroticism react strongly to negative emotions and are less consistent in expression [56,57,58]. These differences could affect how individuals engage in interactions and experience emotions. Future studies could incorporate these individual factors to better understand deeper insights into the transmission of depressive symptoms and physical engagements within organizations. Finally, the workplaces investigated in this study were Japanese companies. Emotions have universal elements across cultures, but their expression and interpretation are more accurate within the same cultural context [59]. These cultural norms could influence the interplay between physical engagement and depressive symptoms during interactions. For example, Japan, known as a high-context culture, tends to rely more on implicit communication [60]. In this setting, nonverbal communications likely play a crucial role in emotional exchanges [61]. Future studies could consider these cultural differences.

## 5. Conclusions

In summary, we investigated the association between physical engagement during face-to-face interactions and depression scores interpersonally associated in the workplace. To examine this, we measured the physical engagement during face-to-face interactions and depression scores of 449 employees from 10 workplaces across various industries. Our findings provide evidence that physical engagement during face-to-face interactions in the workplace is interpersonally associated with employees’ mental health. This suggests that the degree of physical engagement or emotional state is not merely a manifestation of one’s own state but is intertwined with that of surrounding individuals. This interconnectedness can extend beyond the individual level to include interpersonal dynamics and collective implications. Workplace interactions are unavoidable and inevitable and potentially exert a profound influence on employees’ emotional states and professional outcomes. Our study sheds light on the critical significance of interpersonal interactions in relation to the mental health of organizational members.

## Data Availability

The data presented in this study are available on request from the corresponding author. The data are not publicly available due to a confidentiality agreement with the data provider that restricts their use and distribution.

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
