# Peer review of "Physical Engagement in Face-to-Face Interaction Is Associated with Depressive Symptoms of Interaction Partners in the Workplace"

_behavsci, 2024, doi:10.3390/bs14111006_

Round 1

Reviewer 1 Report

Comments and Suggestions for Authors

I read you manuscript and I appreciate your rigor in the statistical treatment. To say something, just two aspects: the use of the term 'correlation' instead of 'relation', as you try to link DS with PE, and, in the conclusions PE with DS (you give son hints later on), and when you report tests on correlation coefficients, you omit to say if you have performed one-tailed or two-sided tests (lines 283 and following), as you conclude about the direction of the correlations (although, the results would be almost the same, but for the p-values in case you report for two-tailed tests).

As the relation between PE and DS is quite low, how can you explain the causes about these relations; are the cases of engagement between employees and their superiors? or another type? Should interactions be a cause of depression?

In the data link, line 511, it is not possible to find the material.

Reviewer 2 Report

Comments and Suggestions for Authors

1.     The literature review lacks depth. It should provide a comprehensive overview of existing studies on emotional contagion, physical engagement, and their implications in workplace settings. More recent studies (post-2020) should be included to ensure relevancy.

2.     Expand on the data pre-processing section. Clearly explain the rationale behind treating interactions within five minutes as continuous and provide quantitative data on the number of interactions affected by this correction. Discuss the potential biases introduced by this method and how they might affect the results.

3.     Justify the choice of CES-D by comparing it to other depression scales (e.g., PHQ-9, Beck Depression Inventory) and highlighting its strengths (e.g., established reliability and validity, suitability for large-scale epidemiological studies). Specify how the CES-D was administered (e.g., online, paper-based, etc.) and mention any measures taken to ensure its reliability and validity in this specific workplace context. Report Cronbach's alpha or other relevant reliability statistics.

4.     Provide more detail on the assumptions of the t-test (normality, independence) and Pearson's correlation (linearity, homoscedasticity). Explain how these assumptions were checked (e.g., visual inspection of histograms, Shapiro-Wilk test, etc.) and any steps taken to address violations if any were found (e.g., transformations).

5.     Clearly state why bootstrap resampling was used (e.g., to obtain more robust confidence intervals, especially given potential non-normality or small sample sizes). Explain how the confidence intervals were calculated and what they represent.

6.     Discuss potential mechanisms for the observed correlation. For example, consider the role of emotional contagion, empathetic responses, or the impact of observing another person's engagement on one's own emotional regulation. Explore whether high physical engagement might act as a buffer against depressive symptoms in partners by fostering positive social interactions.

7.     Clearly define the criteria used to categorize individuals as "high," "middle," and "low" influence. This could involve using statistical methods (e.g., percentiles, standard deviations) to objectively define these groups. Explain the rationale for choosing these specific cut-offs.

8.     Connect the findings to relevant organizational behavior theories, such as social exchange theory, social contagion theory, or emotional labor theory. Discuss how the results support or challenge existing theoretical frameworks and suggest avenues for future research based on these theoretical lenses.

9.     Expand the discussion of individual differences. Suggest specific personality traits (e.g., extraversion, neuroticism) or physiological factors (e.g., cortisol levels) that might moderate the relationship between physical engagement and depressive symptoms.

10.  Expand the limitations section to include the cultural context of the study. Acknowledge that the findings might not be generalizable to other cultures and discuss the potential influence of cultural norms on nonverbal communication and expressions of depression.

Comments on the Quality of English Language

 Minor editing of English language required.

Round 2

Reviewer 2 Report

Comments and Suggestions for Authors

Thank you for making the appropriate corrections to my comments.

The manuscript has been much improved and is in a nice condition now.

I considered that the modifications made improve the quality of the manuscript.